# Enhancing Photocatalytic Activity of ZnO Nanoparticles in a Circulating Fluidized Bed with Plasma Jets

Shiwei Ma [1,†], Yunyun Huang [1,†], Ruoyu Hong [1,*] , Xuesong Lu [2], Jianhua Li [3] and Ying Zheng [1,4]

1   College of Chemical Engineering, Fuzhou University, Fuzhou 350108, China; N180427045@fzu.edu.cn (S.M.); huangyunyun@fzu.edu.cn (Y.H.); xuying31@fzu.edu.cn (Y.Z.)
2   School of Engineering and Physical Sciences, Heriot-Watt University, Edinburgh EH14 4AS, UK; L.Xuesong@hw.ac.uk
3   Coal Chemical R & D Center of Kailuan Group, Tangshan 063611, China; ecljh@kailuan.com.cn
4   Department of Chemical and Biochemical Engineering, Western University, London, ON N6A 5B9, Canada
*   Correspondence: rhong@fzu.edu.cn; Tel.: +86-188-5919-9060
†   These authors contributed equally to this work.

**Abstract:** In this work, zinc oxide (ZnO) nanoparticles were modified in a circulating fluidized bed through argon and hydrogen (Ar–H) alternating-current (AC) arc plasma, which shows the characteristics of nonequilibrium and equilibrium plasma at the same time. In addition, a circulating fluidized bed with two plasma jets was used for cyclic processing. The catalytic degradation performance on Rhodamine B (Rh B) by Ar–H plasma-modified ZnO and pure ZnO was tested in aqueous media to identify the significant role of hydrogen atoms in Rh B degradation mechanism. Meanwhile, the effects of plasma treatment time on the morphology, size and photocatalytic performance of ZnO were also investigated. The results demonstrated that ZnO after 120-min treatment by Ar–H plasma showed Rh B photocatalytic degradation rate of 20 times greater than that of pure ZnO and the reaction follows a first kinetics for the Rh B degradation process. Furthermore, the photocatalyst cycle experiment curve exhibited that the modified ZnO still displays optimum photocatalytic activity after five cycles of experiment. The improvement of photocatalytic activity and luminescence performance attributes to the significant increase in the surface area, and the introduction of hydrogen atoms on the surface also could enhance the time of carrier existence where the hydrogen atoms act as shallow donors.

**Keywords:** plasma; zinc oxide; photocatalysis; nanomaterials





## 1. Introduction

The world environmental problems today are excessive pollution, waste of resources and energy shortages. Semiconductor photocatalysis on waste or pollution treatments is a promising environment-friendly and effective method [1]. This technology makes full use of the semiconductor photocatalysts through photoelectric chemistry to degrade organic pollutant molecules based on the efficient use of solar energy [2,3].

Among various semiconductors, zinc oxide (ZnO) with the wide band gap (3.37 eV) [4], large exciton binding energy (60 meV) [5], good photoelectric properties, nontoxicity, abundance and environmental stability [6,7] performs much better than other semiconductors. Therefore, ZnO has been widely researched for various applications including photo-catalysts [8–10], chemical sensors [11], transparent electrodes [12], solar cells [13] and luminescent materials [14,15].

In the past decades, the application of ZnO in the field of photocatalysis has gradually become known since the Honda-Fujishima effect was reported in 1970s. In the photocatalytic process, ZnO nanoparticles are not easy to react with other substances and resistant to high temperatures. At the same time, when ZnO is irradiated by ultraviolet light, the electrons obtain light energy to transit from the valence band to the conduction band, and generate electron–hole pairs. Simultaneously, some electrons return to the valence

band in the form of heat and light emission, and other carriers move on the ZnO surface. Among them, electrons have strong reducing ability, and photogenerated holes have strong oxidizing properties. They further react with pollutants to achieve photocatalytic effects [16].

However, ZnO nanoparticles as photocatalyst also suffers several problems: (i) The bandgap of ZnO is too wide for electron transition, which makes it only respond to the ultraviolet region of sunlight [8], and the ultraviolet region only occupies a ratio of 5–7% sunlight. To expand the region of light response, modification of ZnO nanopowders is essential. (ii) The rapid recombination of electron–hole pairs in the catalytic process is also the main problem that seriously affects the photocatalytic performance [17–19]. To improve the photocatalytic performance of ZnO, the reduction in the energy band gap of ZnO and the effective separation of the photogenerated carriers have to be achieved. Several methods for modifying ZnO have been developed, such as metal element doping modification [20], nonmetal element doping modification [21], semiconductor material composite loading [17,22], surface modification [23] and so on. However, these methods have disadvantages such as cumbersome preparation, serious post-treatment process or serious pollution of by-products, which limits the large-scale application of these methods [24].

Currently, plasma become a research hot point and has been extensively used in synthesis and modification of ZnO in recent years [25–27] because the high energy of the plasma could remove the surface state of the material, surface impurities or defects, and different plasmas would produce doping, deposition or reaction phenomena [26]. In particular, the arc plasma is highly valued due to combining the characteristics of nonequilibrium plasma and equilibrium plasma. Continuous modification of materials by arc plasma is a facile method of high yield [28].

Photocatalytic performance of ZnO depends on the modification conditions and methods [29]. Dao et al. [30] modified ZnO thin films by Ar plasma, and etching was observed on the surface. The grains on the surface were etched out, leading to a flatter surface with a smaller roughness. It suggested that the improvement of photoelectronic properties may be due to the effects of hydrogen ions produced by high-energy plasma ionized residual gases. Dev et al. [23] and Baratto et al. [31] reported similar improvements by Ar plasma treatment and proposed that the effects were attributed to incorporation of hydrogen. Nam et al. [26] proposed the synthesis and modification methods of ZnO nanoparticles. No changes in specific surface area were observed, but oxygen was introduced into the ZnO surface and O-H stretching peak was increased on the ZnO nanoparticles surface. The increase in free radicals is the main reason to enhance the photocatalytic performance. Nitrogen ($N_2$) [32] and ammonia ($NH_3$) [33] plasma treatments were also reported, and they showed similar effect of the argon plasma.

In the previous work by research groups, the preparation method of zinc oxide [34] was studied, and modification methods such as aluminium doping [35] and Ar plasma [36] were also studied. In this article, based on previous research work, we combined plasma and fluidized bed to develop a more efficient, green and convenient new modification method.

In this study, ZnO nanoparticles were modified by Ar and Ar–H plasma in a circulating fluidized bed, and factors on the photocatalytic performance were studied including plasma treatment time. We compared the photocatalytic activity of ZnO before and after modification and studied the mechanism of plasma to improve the photocatalytic performance. At the same time, the role of hydrogen ions in modification process was investigated. Finally, the ZnO nanoparticles with excellent photocatalytic ability were successfully obtained and applied.

## 2. Results

### 2.1. XRD Analysis of ZnO before and after Modification

Figure 1 shows the XRD patterns of ZnO before and after modification. The characteristic diffraction peaks of the samples are 31.7°, 34.4°, 36.2°, 47.5°, 56.7°, 63.0°, 66.4°, 68.1° and 69.3°, corresponding to the crystal planes of (100), (002), (101), (102), (110), (103),

(200), (112) and (201), respectively. This result is consistent with the diffraction peaks of hexagonal wurtzite phase ZnO in the JCPDS standard card (No. 361451) [37]. As shown in Figure 1, the width at half maximum (FWHM) of ZnO before and after plasma treatment are 0.52962 and 0.45477, respectively, suggesting that the plasma treatment increased the crystallite size of ZnO nanoparticles and the plasma modification process did not just act on the surface.

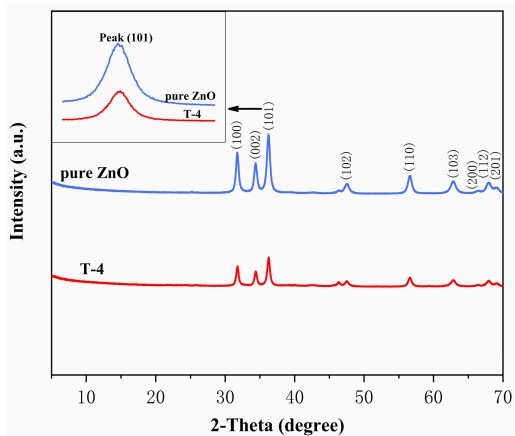

**Figure 1.** XRD patterns of pure ZnO nanoparticles and T-4 sample.

Figure 2 shows (002) and (100) XRD profiles of ZnO nanoparticles before and after 120 min treatment. The (002) peak moved to a lower angle, while the (100) peak moved to a higher angle, corresponding to an increase in the lattice constant *c* and a decrease in the lattice constant *a*, respectively. These results mean that new ions/atoms are introduced at the crystal plane through plasma treatment and modification, and the lattice united in the crystal grain block are stressed and deformed. At the same time, it can also be proved that plasma acts not only on the surface but also on the bulk crystalline structure.

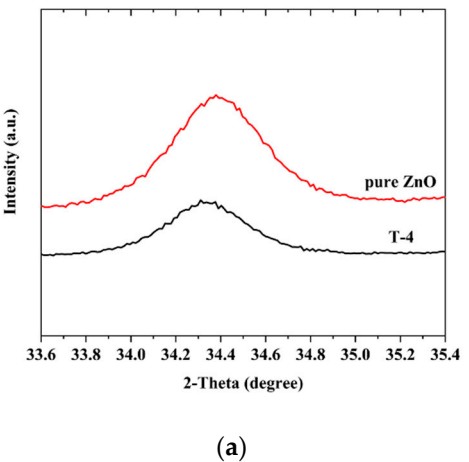

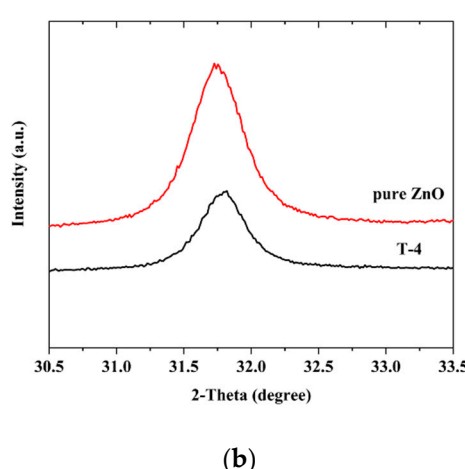

(**a**)                                         (**b**)

**Figure 2.** Effect of the Ar–H plasma treatment time on the structural properties of the ZnO nanoparticles. (**a**) Out-of-plane 002 and (**b**) in-plane 100 XRD.

## 2.2. SEM Analysis of ZnO before and after Modification

SEM images of original ZnO and T-4 samples are shown in Figure 3a,b, respectively. As shown in Figure 3b, the pure ZnO shows a spherical agglomerated structure with a diameter of 100–800 nm. After 120 min of modification process, the etching and bombardment effects were found on the surface of ZnO as well as a decrease in the degree of agglomeration, accompanied by smoother surfaces and smaller roughness, as shown in Figure 4a. In theory,

the improvement of dispersibility and the reduction in nanoparticle size can effectively increase the specific surface area of ZnO, thereby increasing the number of reactive sites and improving photocatalytic efficiency. The specific verification can be observed from the BET test below.

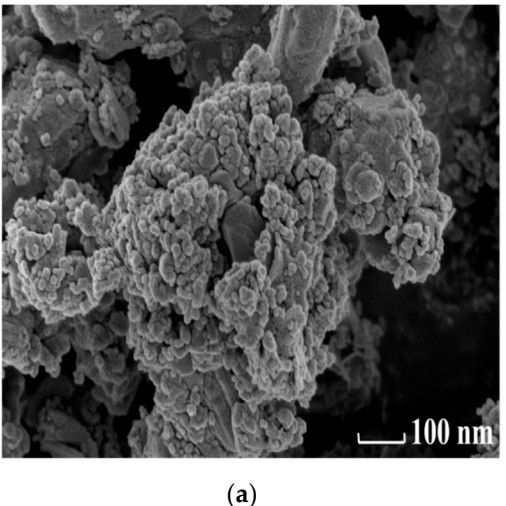

(a)

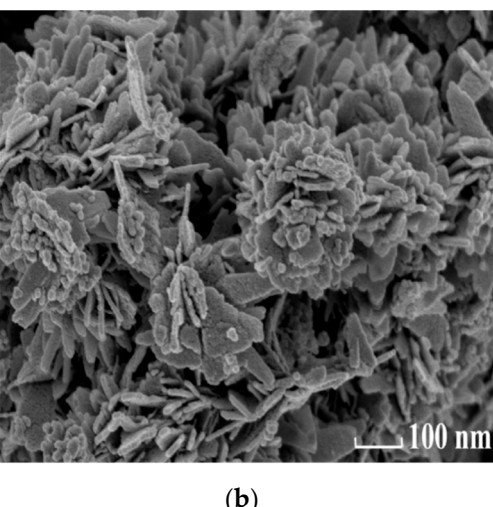

(b)

**Figure 3.** SEM images of products in different time: (**a**) 120 min. plasma treatment (T-4 sample) and (**b**) pure ZnO.

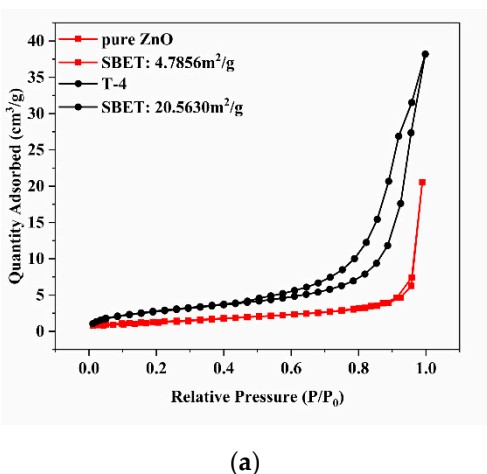

(a)

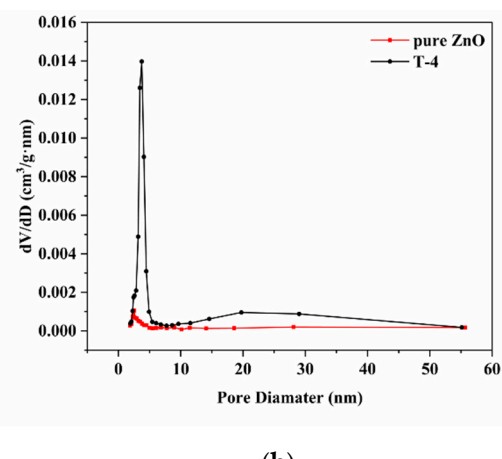

(b)

**Figure 4.** (**a**) Nitrogen adsorption-desorption isotherms and (**b**) pore size distribution curves.

### 2.3. Characterization of Specific Surface Area and Carrier Lifetime

Figure 4 shows the changes in specific surface area and pore size distribution before and after plasma modification of ZnO. As shown in Figure 4a, the specific surface area of ZnO increased to 20.5630 m$^2$/g after 120 min of plasma treatment, which is 4.3 times higher than that of pure ZnO. This may be due to the high energy of the plasma particles that etch the ZnO surface. The increase in the specific surface area can provide more reaction active sites for the photocatalytic reaction and improve the degradation efficiency. Figure 4b shows the pore size distribution curve of the sample. It can be seen that the small pores of pure ZnO and T-4 are all distributed around 3 nm, and the large pores of T-4 are all distributed between 15 and 30 nm. This porous structure is possibly caused by the gaps between small nanoparticles.

Photoluminescence spectrum (PL) was used to examine the luminescence intensity of ZnO and the recombination time of photogenerated electrons and holes. As shown in Figure 5a, after plasma treatment, the luminescence intensity of ZnO at 380 nm decreased

significantly, but the luminescence intensity increased at 450 nm. This may be due to the following two reasons:

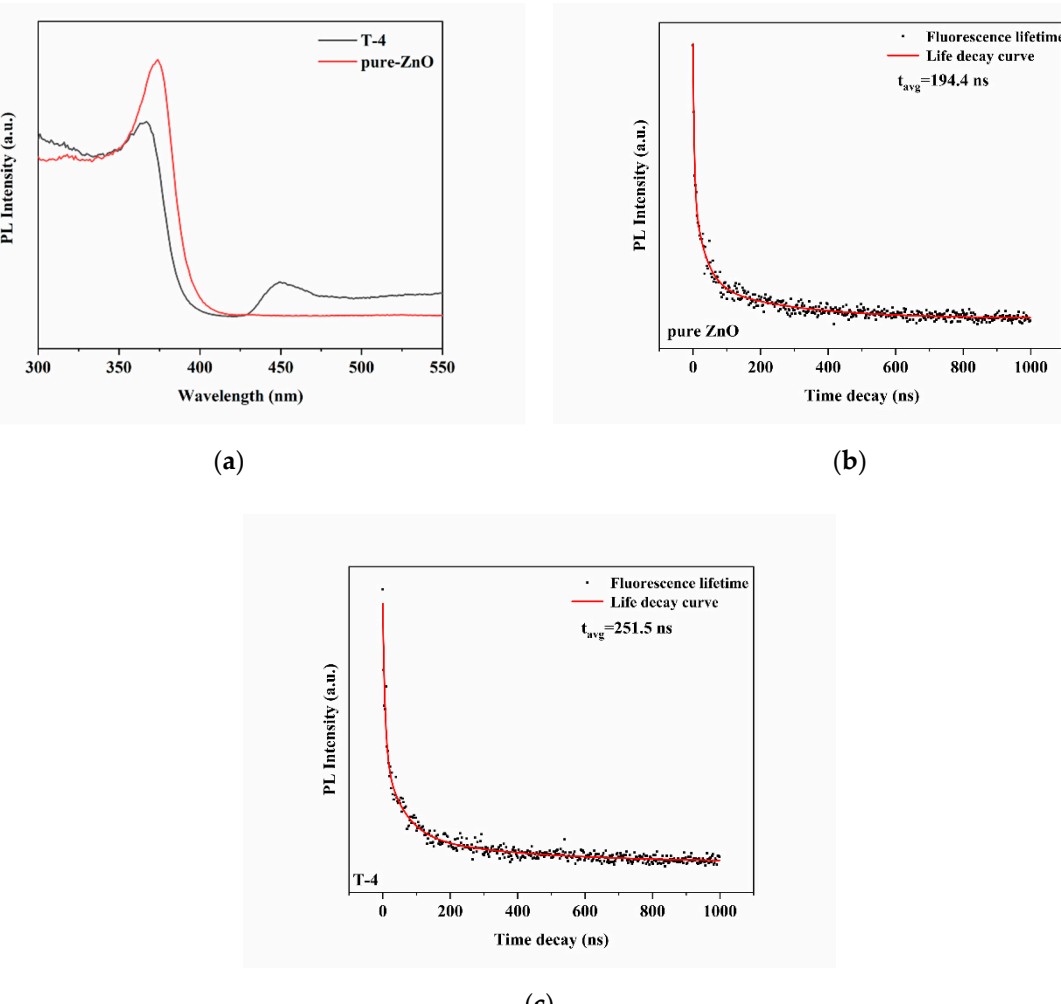

(a)

(b)

(c)

**Figure 5.** (**a**) Photoluminescence spectra of ZnO and T-4, (**b**) Carrier lifetime curve of pure ZnO and (**c**) carrier lifetime curve of T-4 sample.

(i) The recombination rate of photogenerated electrons and holes decreases, which leads to a decrease in luminescence intensity at 380 nm [38] and (ii) the slight increase in luminescence intensity at 450 nm may be related to the increase in oxygen vacancy concentration [39], while the reason why this peak is not observed in pure zinc oxide may be due to the too low concentration of oxygen vacancies.

The existence of average lifetime was also found in the decay profile at 375 nm in Figure 5b,c [40]. The carrier average lifetime of ZnO was changed from 194.4 to 251.5 ns, suggesting that the photogenerated electron–hole recombination rate slowed down, which also proved the above conclusion.

### 2.4. XPS Analysis

X-ray electron spectroscopy (XPS) was used to eValuate the difference before and after Ar–H plasma surface treatment. Figure 6a shows the XPS spectra for Zn 2p core level peaks; the Zn 2p core energy peak shifted from 1022.6 eV to the binding energy region of 1021 eV. The results display that the Zn on the surface of ZnO changes from oxygen-rich zinc before treatment to ZnO containing metallic zinc after treatment. O 1s spectra is shown in Figure 6b; it can be seen that the main O 1s peaks of ZnO nanoparticles are fitted to the three peaks of 531.25, 532.7, and 534.4 eV, corresponding to Zn-O, the physically

absorbed OH radicals and molecular water on the surface of ZnO, respectively [41]. The shoulder peaks matching the OH radicals and molecular water increase significantly after plasma treatment and the peak intensity increased significantly, indicating that the chemical adsorption and physical adsorption of OH radicals and molecular water on the surface of ZnO are enhanced. Figure 6c shows the specific changes in the intensity of OH radicals and molecular water peaks before and after plasma treatment. The results show that Ar–H plasma treatment not only increases the concentration of OH radicals but also has an effect on the surface adsorption of molecular water in the air, which can cleave the double bond (C=N) to decompose the Rh B solution and couple with the N-H single bond of the Rh B solution, improving effectively the photocatalytic effect ultimately.

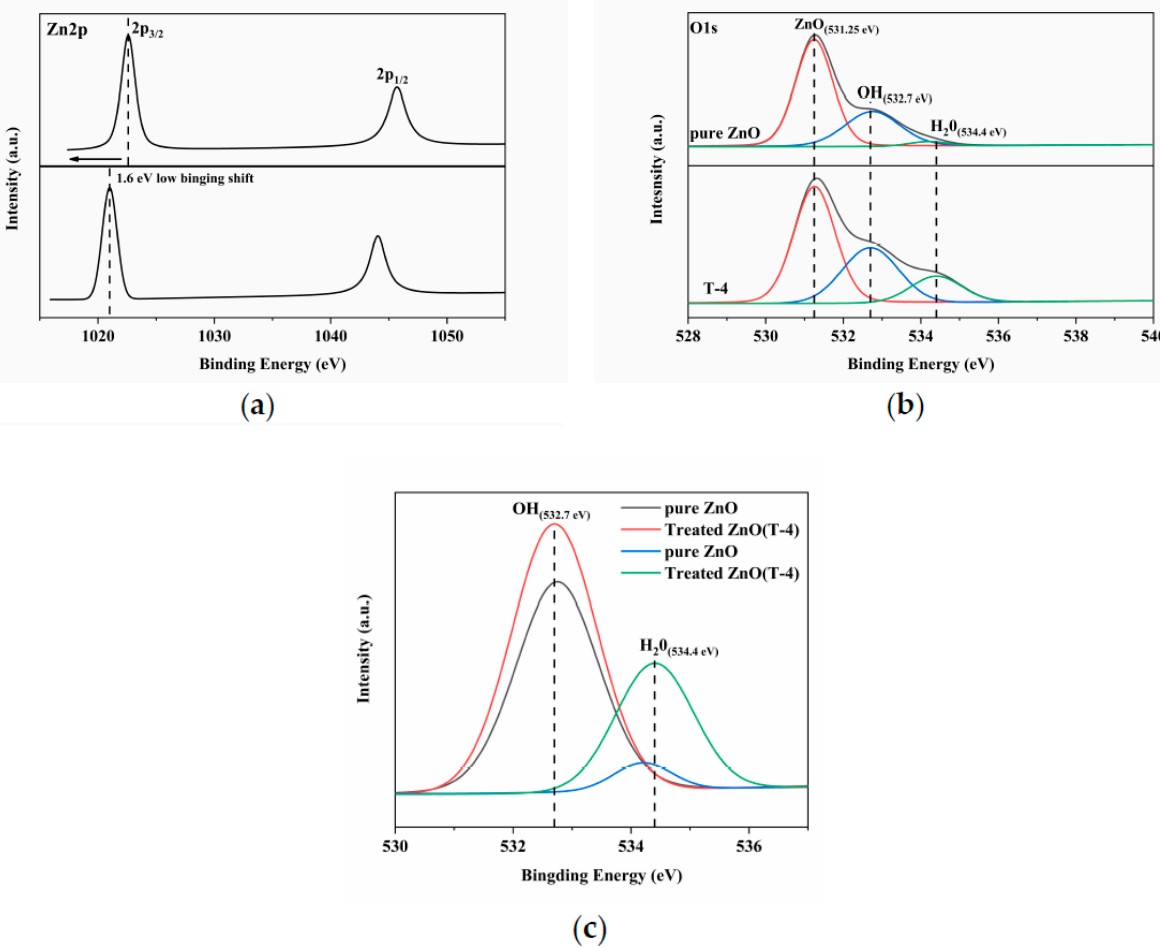

**Figure 6.** Changes in XPS spectra of ZnO in the (**a**) Zn 2p, (**b**) O 1s peak and (**c**) OH and $H_2O$ peaks in O 1s regions before and after Ar–H plasma treatment.

### 2.5. Photocatalytic Degradation of Rh B Solution

To study the effect of treatment time on the activity of semiconductor catalysts, ZnO (T-1, T-2, T-3, T-4 and T-5) was used for photocatalytic degradation of Rh B solution. The experimentally measured change curve of $C/C_0$ of Rh B solution with catalytic time is shown in Figure 7a. The blank experiment is the Rh B solution, which is directly illuminated by a 50 W light source with no catalyst. It is worth noting that ultraviolet light illumination in the absence of any photocatalyst or dark conditions with catalysts do not lead to the degradation of Rh B. In fact, in the blank experiment, only 6.3% of the Rh B solution was degraded under the UV light source within 100 min (6.3%). In the degradation experiment after addition of pure ZnO, the degradation rate of Rh B was improved but still showed

poor degradation capacity. The Ar–H plasma-modified ZnO significantly enhanced the degradation of Rh B, and the degradation rate first increased and then decreased with time. The degradation efficiency was the highest when the plasma treatment is 120 min, and the Rh B solution was degraded by 97.8% within 40 min.

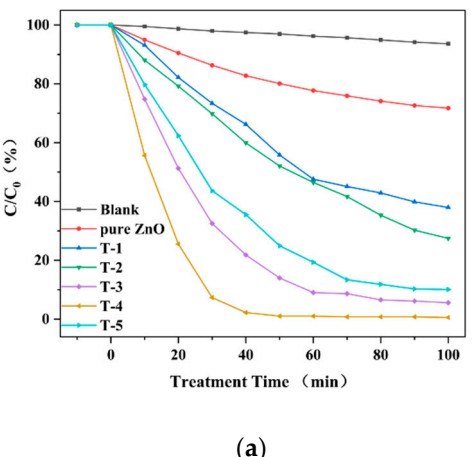 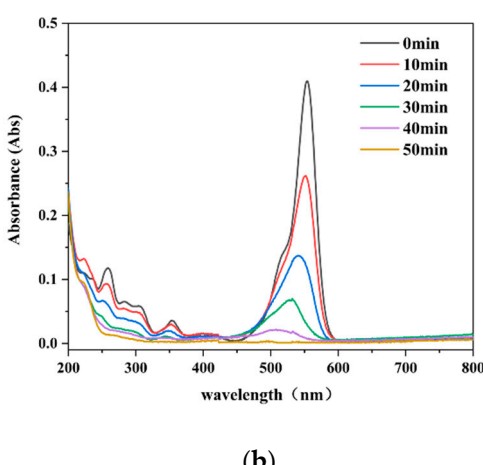

| (a) | (b) |
|:---:|:---:|

**Figure 7.** (**a**) The Rhodamine B (Rh B) concentration of different samples changes with time. (**b**) The absorbance curve change with time when using T-4 product to degrade Rh B.

The apparent enhancement in photocatalytic degradation capacity of ZnO nanoparticles could be due to the effect of the increase in specific surface area (BET), decrease in electron–hole recombination rate (PL), increase in surface OH radicals (XPS), and H atoms/ions as a shallow donor. However, long-term plasma treatment time will cause the photocatalytic ability to decrease (T-5), which may be attributed to the high energy of the plasma causing irreversible structural damage to the ZnO nanoparticles. In conclusion, the best modification time for plasma is 120 min. The temporal eVolution about the degradation of Rh B solution over T-4 is shown in Figure 7b. The concentration of Rh B solution was indicated by the decrease in the intensity of the absorption peak.

### 2.6. Photocatalytic Degradation Kinetics

The concentration of the solution can be calculated by measuring the absorbance of the Rh B solution, according to the degradation efficiency of each catalyst based on the solution concentration. It is known through Equation (1) fitting that the Rh B degradation process conforms to the first-order reaction kinetics:

$$
\begin{aligned}
-dC/dt &= k \times C \\
-\ln(C/C_0) &= k \times t \\
\ln(C_0/C) &= k \times t
\end{aligned}
\tag{1}
$$

where, $C$, $C_0$, $k$ and $t$ represents the Rh B concentration of "$t$" time, the initial concentration of the dye, reaction rate constant (min$^{-1}$) the time at which photocatalytic degradation takes place, respectively.

Fitting the reaction kinetics of the results of each group of photocatalytic degradation experiments, the results are shown in Figure 8. The degradation rate of Rh B solution decreased significantly after 90% degradation, therefore, for this experiment, a fitting analysis was performed up to a 90% degree of degradation. It can be seen from the Figure 8 that the reaction rate constant $k$ in the blank experiment was only 0.00065 min$^{-1}$. After the catalyst was added, the degradation rate of Rh B was significantly increased, in which the reaction rate constant was 0.08933 min$^{-1}$ when the treatment time was 120 min. The fitting data of the experimental results are shown in Table 1. In all the results, $R^2$ is greater than 0.99, indicating that $\ln(C_0/C)$ in the Rh B degradation experiment results shows a linear relationship with

time *t*, which is a first-order reaction and meet the above equation. It can be seen that the sample with the best photocatalytic degradation effect on Rh B solution is T-4.

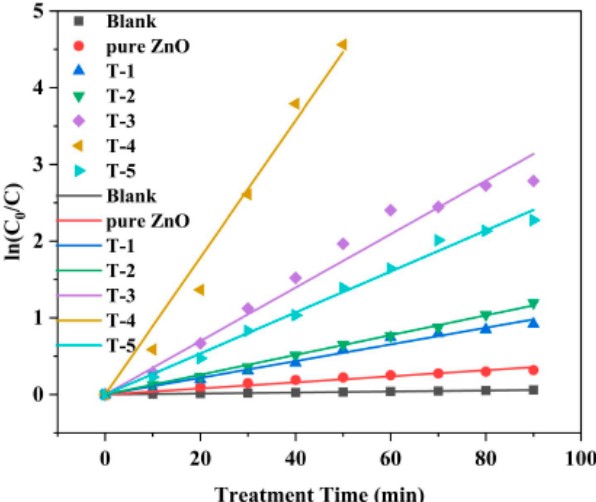

**Figure 8.** The first-order reaction kinetics fitting curve of the degradation of Rh B in T4 sample.

**Table 1.** Degradation effect of samples with different plasma treatment time on Rhodamine B (Rh B) solution.

| Sample | Degradation Rate (%) | Reaction Rate Constant (min$^{-1}$) | $R^2$ |
|--------|---------------------|-------------------------------------|-------|
| Blank | 6.4 | 0.00065 | 0.999 |
| Pure ZnO | 28.3 | 0.00396 | 0.998 |
| T-1 | 62.0 | 0.01089 | 0.994 |
| T-2 | 72.5 | 0.01291 | 0.999 |
| T-3 | 94.4 | 0.03485 | 0.991 |
| T-4 | 99.4 | 0.08933 | 0.991 |
| T-5 | 89.9 | 0.02675 | 0.997 |

To further study the mechanism of improving photocatalysis and the mechanism of plasma modification, more characterization methods were used. The ultraviolet–visible diffuse reflectance spectroscopy (DRS) was used to characterize the degree of electron–hole pairs generated by the samples and the band gap width. As shown in Figure 9a, the ultraviolet–visible (UV) absorption edge wavelength was observed at the wavelength of 375 nm, which is the inherent absorption band gap of ZnO (3.2 eV) [42]. The plasma modification process did not affect the change of the main absorption band of ZnO, the main absorption band of ZnO is shorter than 400 nm, and the absorption band of the T-4 sample was very similar to that of pure ZnO. T-4 sample also reveals other absorption peaks, which is due to plasma treatment in the visible light region (400–700 nm). T-4 nanomaterial exhibits an absorption ability also in the visible range.

The energy band theory provides a theoretical basis for the catalytic process of ZnO. As a typical semiconductor, ZnO exhibits a valence band and a conduction band. The valence band is usually occupied by electrons and exhibits lower energy, while the conduction band is empty and the energy is higher than the valence band. After the electron gains enough energy, it will transition from the valence band to the conduction band and generate carriers. The minimum energy required for the transition is $E_g$ [43].

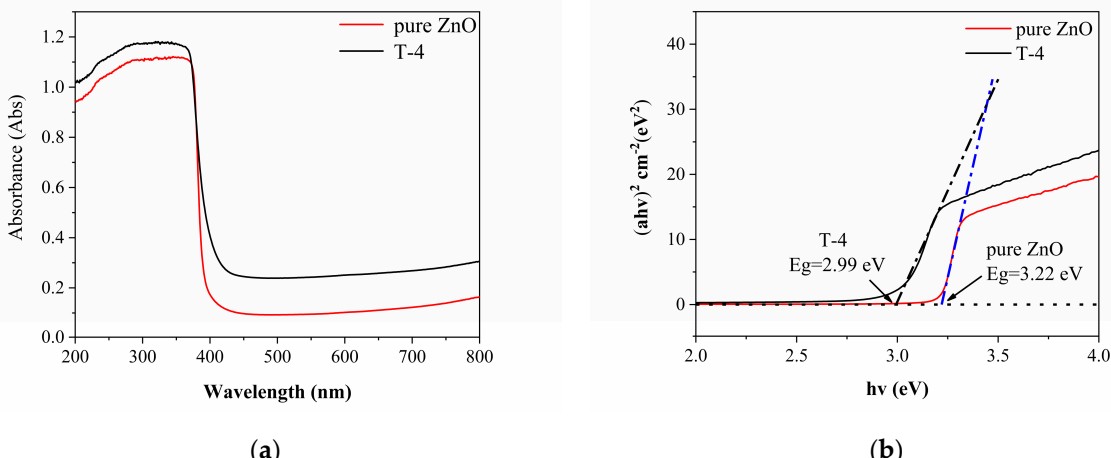

**Figure 9.** (**a**) Ultraviolet–visible absorption spectrum and (**b**) the corresponding fitting curves and the fitting result of the band gap.

According to the Kubelka-Munk function [44], the band gap ($E_g$) is determined from the absorption spectra using equation as follows:

$$\alpha h v = A(h v - E_g)^n, \tag{2}$$

where, $\alpha, h, v, A, E_g, \eta$ represents optical absorption coefficient, Planck constant, frequency of the incident photon, absorption constant for direct transition, band gap and index characterized by light absorption process, respectively. ZnO is a direct band gap semiconductor, so the value of $\eta$ is 1/2. The values of $\alpha$ and $A$ can be obtained according to the Formula (3) and (4):

$$\alpha = A(1-R)^2/2R \tag{3}$$

$$A = -\lg(R), \tag{4}$$

where, $R$ is the reflectance. As shown in Figure 9b, the band gap was calculated by Equations (2)–(4). The results showed that the band gap of ZnO was reduced from 3.22 to 2.99 eV after 120 min plasma modification process. Theoretically, the narrower the band gap, the easier it is to generate carriers, which has a higher photocatalytic degradation efficiency. The possible reason is that hydrogen acts as a shallow donor in ZnO, leading to a reduced band gap [30].

### 2.7. Cycle Experiment

The recycling of photocatalyst is of great significance to practical applications. When evaluating the performance of a catalyst, stability is also an important indicator. The less the catalytic efficiency decreases after the catalyst is recycled, the stronger the stability. Through five repeated experiments under the same conditions, the stability of T-4 sample to the photocatalytic degradation of Rh B solution was tested. As shown in Figure 10a,b, the T-4 sample degraded about 99.4% of Rh B in the first circle and the degradation efficiency of Rh B decreased to 89.76% within 100 min in the 5th cycle test. The decrease in photocatalytic efficiency may be due to the scattering of ionized impurities during the cycle. In addition, the process of centrifugation, washing and drying will also cause sample loss, so it is reasonable to maintain a degradation rate close to 90% after 5 cycles.

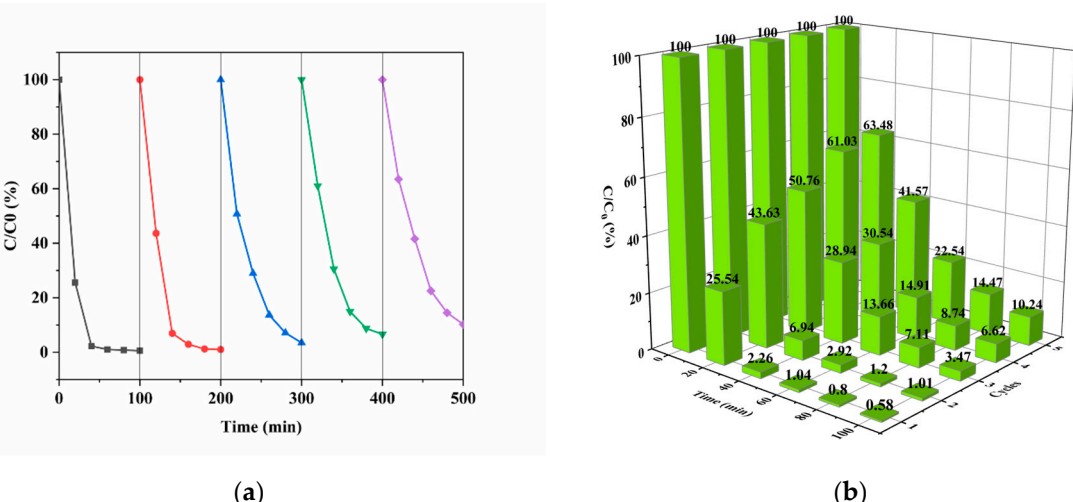

(**a**)  (**b**)

**Figure 10.** (**a**) Cyclic degradation curve and (**b**) three-dimensional curve of Rh B.

## 3. Discussion

Based on the above analysis, the mechanism to improve the photocatalytic activity was speculated. The possible mechanism responsible for the increases in photocatalytic activity is due to oxygen vacancies produced by the etching of argon-hydrogen plasma [45–47]. Oxygen vacancies are considered to be one of the main defects generated by carriers in ZnO, and the increase in oxygen vacancies concentration will increase the photocatalytic activity of ZnO [48]. First, the plasma particles have ultrahigh energy, which can modify the surface of ZnO to produce some defects without changing the lattice structure. Moreover, hydrogen plasma can combine with oxygen to deepen the concentration of oxygen vacancies, theoretically. Another mechanism that leads to the improvement of photocatalytic performance may result from hydrogen or hydrogen incorporation acting as shallow donors. As shown in Figure 11, hydrogen may also occupy interstitial positions in the ZnO lattice and form impurity levels, which helps to increase the carrier concentration. In addition, we also discussed the problem of photocatalytic performance degradation caused by prolonged plasma modification. One possible explanation is that the ionized impurities are scattered, when the modification time is extended and the concentration of hydrogen ions reach a certain value, hydrogen will help the charge scattering [49,50]. On the other hand, prolonged plasma modification time will damage the surface of the sample, generate unsaturated dangling bonds and capture photogenerated electrons or holes [51]. These factors ultimately reduce the number of reactive sites and the concentration of surface free radicals.

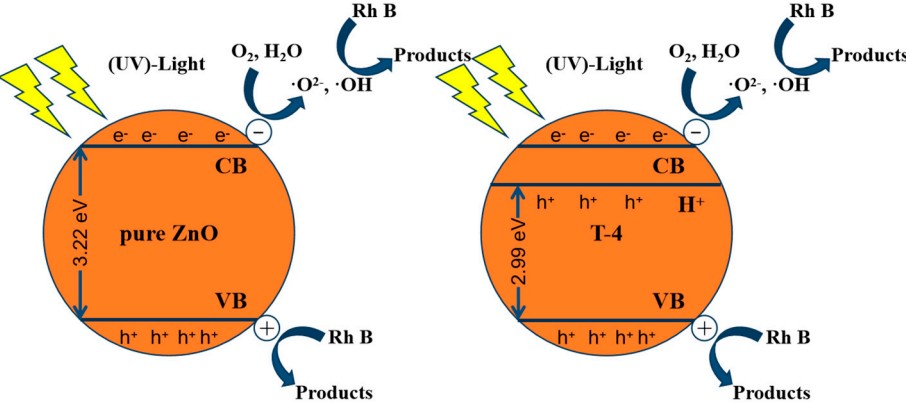

**Figure 11.** Mechanism analysis of pure ZnO and T-4 sample.

## 4. Materials and Methods

### 4.1. Materials and Apparatus

Argon (99.99%) and hydrogen (99.99%) were purchased from Xinhang Industrial Gases Co. Ltd. (Fuzhou, China). AC high voltage power supply and transformer were obtained from Jiaxing Jialin Electronic Technology Co. Ltd. (open circuit voltage 20 kV, Jiaxing, China). Zinc oxide (ZnO) nanoparticles were obtained from Aladdin Co. (Shanghai, China).

Figure 12a shows plasma fluidized bed equipment used in the experiments, which is made up of stainless steel. It mainly consists of two compartments: (i) The plasma generator is composed of a pair of ceramic discharge nozzles (length 150 mm and inner diameter 40 mm) and two pairs of parallel electrodes (length 120 mm and bottom diameter 20 mm) and (ii) The separation and collection units include cyclone separation, bag filter and induced draft fan. The fan helps the carrier gas to circulate the ZnO in the plasma reactor. The products are mainly collected by a cyclone separator, and the rest is collected after being separated from the exhaust gas through a cloth bag. The details on the facility are referred to Figure 12a and Table 2 [52].

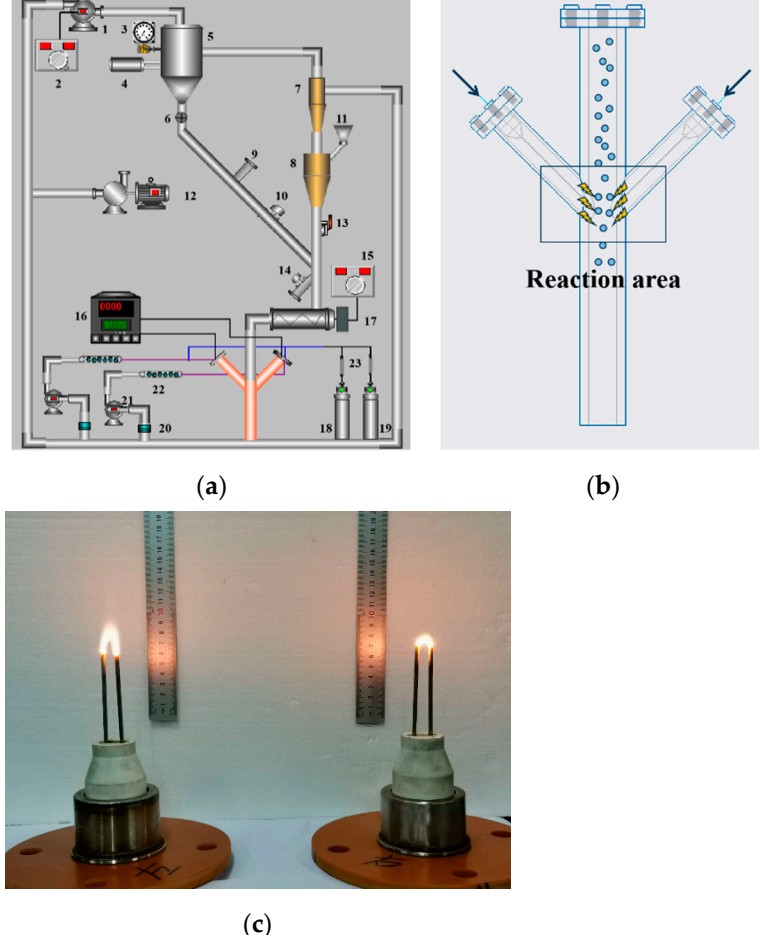

(a)

(b)

(c)

**Figure 12.** (**a**) Schematic diagram of modification process device, (**b**) specific reaction area, and (**c**) the plane of the AC sliding arc that occurs on the top of the flat electrode.

**Table 2.** Detailed parts list.

| Part Number | Part Name | Part Number | Part Name |
|---|---|---|---|
| 1 | Induced draft fan | 12 | Vacuum pump |
| 2 | Governor-1 | 13 | Valve-2 |
| 3 | Pressure gauge | 14 | Outlet |
| 4 | Blower | 15 | Governor-2 |
| 5 | Bag separator | 16 | AC power |
| 6 | Rotary feeding valve | 17 | Screw propeller |
| 7 | Cyclone separator-1 | 18 | Argon |
| 8 | Cyclone separator-2 | 19 | Hydrogen |
| 9 | Oscillator | 20 | Filter membrane |
| 10 | Valve-1 | 21 | Circulation pump |
| 11 | Feeding port | 22 | Desiccant |
| The diameter of all pipes in the device is 600 mm | | 23 | Flowmeter |

### 4.2. Modification Process of ZnO

The modification process of ZnO nanoparticles by argon–hydrogen (Ar–H) arc plasma is shown in Figure 12a and the specific reaction area is shown in Figure 12b. First, the vacuum pump extracts the air from the reaction device to −0.1 MPa. Then, the induced draft fan was turned on to induce the Ar–H mixed gas into fluidized bed. When the AC power was supplied, the arc was generated between two parallel electrodes at the top region (as shown in the Figure 12c). The arc discharged in the direction of the airflow in a yellow white colour. With the Ar–H gas flow rate increasing, the semicircular plasma beam enlarged and tended to be flame shape. At the same time, the semicircular plasma beam became sparse and nonuniform as the arc was elongated. The inhomogeneous arc was probably owing to the rough surfaces of the parallel electrodes. Finally, when the arc remained stable, Ar and $H_2$ gas mixture was introduced in proportion into the plasma reactor where the ratio and the gas flow rate were controlled and adjusted by a flow meter. The plasma treatment time of ZnO were kept constant in five groups, which were 30, 60, 90, 120, and 150 min, and the corresponding products were denoted as T-1, T-2, T-3, T-4 and T-5, respectively.

### 4.3. Characterizations

The crystal structure of ZnO nanoparticles was investigated by X-ray diffraction (XRD) and the data were collected on Cu K$\alpha$ radiation (DY1602/Empyrean, Malvern Panalytical, Malvern, UK), using a step size of 0.2° and a counting time of 1 s per step in the range of 5–80°. The surface morphology of plasma-treated ZnO nanoparticles was characterized by scanning electron microscopy (SEM, TecnaiG220, FEI, Hillsboro, OR, USA). PL spectra was used to observe luminous performance at room temperature. The specific surface area was measured by Brunauer–Emmett–Teller (BET, ASAP 2460, Mike Company, Hong Kong, China). To investigate the photocatalytic capacity and optical performance of ZnO, the measurement by ultraviolet–visible spectroscopy (UV–VIS, TU-1900, Beijing Puxi General Instrument Co., Ltd., Beijing, China) was carried out in the range of 200–900 nm. The band gap and the photocatalytic reaction kinetic constant are calculated by UV diffuse reflectance, absorption spectra using the Kubelka-Munk method and the degradation curve of Rhodamine B. X-ray photoelectron spectroscopy (XPS, ESCALAB 250, Thermo Fisher Scientific, Waltham, MA, USA) was used to characterize element differences before and after plasma treatment. The average lifetime was calculated by Formula (5) [53].

$$\tau avg = (\alpha_1 \tau_{12} + \alpha_2 \tau_{22} + \alpha_3 \tau_{32})/(\alpha_1 \tau_1 + \alpha_2 \tau_2 + \alpha_3 \tau_3), \tag{5}$$

where $\tau avg$ is average lifetime, $\tau_1$, $\tau_2$, and $\tau_3$ are decay times and $\alpha_1$, $\alpha_2$, and $\alpha_3$ are relative magnitudes [54].

### 4.4. Measurement of Photocatalytic Abilities

The photocatalytic capacity of plasma-modified ZnO nanoparticles was investigated by degrading Rhodamine B (Rh B). In the experiments, 0.1 g of the modified samples was dispersed in 200 mL of Rh B aqueous solution (10 mM). The suspension was kept in dark condition for 60 min until reaching adsorption and desorption equilibrium before illumination. Subsequently, the reaction was carried out at room temperature under a 50 W high-pressure mercury lamp (365 nm) with continuous cooling water. Before the irradiation, 4 mL of dye mixture was taken to centrifuge for eVery 10 min, and then the UV–VIS spectrophotometer was used to analyse supernatant at 554 nm. Therefore, the photocatalytic degradation curves of Rh B solution were obtained, and the reaction type and kinetic constant of the degradation reaction of rhodamine B were calculated. The degradation percentages of Rh B solution were calculated using Equation (5). The stability of the catalyst was also tested using cycle experiments.

## 5. Conclusions

In this work, a new process combining plasma and fluidized bed has been developed for the modification of ZnO nanoparticles continuously using AC arc plasma. Compared with the traditional plasma modification methods, the process has the advantages of simple operation and continuous modification in large quantities. Under the optimal conditions of plasma modification for 120 min, the band gap of ZnO was reduced to 2.99 eV, and the specific surface area was increased to 20.5630 $m^2/g$. As the plasma treatment time increased, the photocatalytic efficiency of the sample first increased and then decreased, and the maximum degradation rate of Rh B solution was 0.08933 $min^{-1}$, which was 22 times than that of unmodified ZnO. In addition, taking into consideration the characterization results of SEM, XPS, UV–VIS, DRS and other techniques, the photocatalytic mechanism of modified ZnO was speculated. The Ar–H plasma treatment increased the specific surface area while reducing the band gap of ZnO, and an impurity level was formed in the band gap of ZnO by hydrogen elements, which was conducive to generate and transform photogenerated electron–hole pairs. Therefore, the plasma fluidized bed-modified ZnO nanoparticles might provide a new idea to improve the application of ZnO photocatalysis.

**Author Contributions:** Conceptualization, S.M. and R.H.; methodology, S.M.; software, S.M.; validation, R.H., Y.H., J.L. and Y.Z.; formal analysis, S.M.; investigation, S.M.; resources, S.M.; data curation, S.M.; writing—original draft preparation, S.M.; writing—review and editing, J.L., X.L. and Y.Z.; visualization, S.M.; supervision, Y.H.; project administration, R.H.; funding acquisition, R.H. and Y.H. All authors have read and agreed to the published version of the manuscript.

**Funding:** National Natural Science Foundation of China (NSFC, No. 21246002), Minjiang Scholarship of Fujian Province (No. Min-Gaojiao[2010]-117), Central Government-Guided Fund for Local Economic Development (No. 830170778), R&D Fund for Strategic Emerging Industry of Fujian Province (No. 82918001) and International Cooperation Project of Fujian Science and Technology Department (No. 830170771).

**Institutional Review Board Statement:** "Not applicable" for studies not involving humans or animals.

**Informed Consent Statement:** Informed consent was obtained from all subjects involved in the study.

**Data Availability Statement:** Data is contained within the article.

**Acknowledgments:** This research was financially supported by the National Natural Science Foundation of China (NSFC, No. 21246002), Minjiang Scholarship of Fujian Province (No. Min-Gaojiao[2010]-117), Central Government-Guided Fund for Local Economic Development (No. 830170778), R&D Fund for Strategic Emerging Industry of Fujian Province (No. 82918001) and International Cooperation Project of Fujian Science and Technology Department (No. 830170771).

**Conflicts of Interest:** The authors declare no conflict of interest and all the authors approved the manuscript for publication. The funders had no role in the design of the study; in the collection, analyses, or interpretation of data; in the writing of the manuscript; or in the decision to publish the results.

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
