# Peer review of "Enhancing Photocatalytic Activity of ZnO Nanoparticles in a Circulating Fluidized Bed with Plasma Jets"

_catalysts, doi:10.3390/catal11010077_

Round 1
Reviewer 1 Report
The manuscript submitted to Catalysts entitled "Enhancing Photocatalytic Activity of ZnO Nanoparticles in a Circulating Fluidized Bed with Plasma Jets" by Ma and co-workers presents the preparation and characterization of ZnO nanoparticles and their application in rhodamine B photodegradation. The overall subject is interesting and zinc oxide (ZnO) nanoparticles have emerged in recent years as a cost effective route for the treatment of organic pollutants and the transformation of hazardous substances into benign forms and the results of the current study, are in general interesting.
However, a few questions can be raised from the way the authors introduce and discuss the subject. For instance:
a) The introduction seems confusing and it does not highlight the subject properly. It should be clarified why ZnO nanoparticles are a subject of interest, and for the average reader this is not clear. Furthermore, there some typos and phrases that do not add meaningful information to the introduction (e.g. line 38: "Especially, ZnO performs well on the photocatalytic.").
Also, there are missing some references in the first two sentences of the introduction. I feel this subject has been discussed elsewhere and these references should be used at the end of these sentences.
b) In figure 2 the authors compare the intensity of the peaks on the DRX pattern of ZnO and T4 samples and this led to the conclusion that there is less crystallinity in sample T4. If I am not mistaken the authors should compare the peak full width at half maximum of each material in order to draw any conclusion about crystallinity. This is not clear in the text neither in the figure. Perhaps the authors should provide an inset in this figure with an expansion of one peak for comparison.
c) In the catalytic studies the authors show that material T5 shows less photocatalytic performance against rhodamine B when compared with material T4. However, no characterisation (PDRX, SEM, N2 isotherms… etc) of material T5 is provided by the authors. Somewhere between material T4 and T5 there is a threshold where the plasma treatment changes completely the structure-affinity of these materials. In this regard, it would add value if this characterisation was available in the manuscript, since there should be major structural differences between materials T4 and T5.
d) The degradation results with ZnO materials should be compared against the most common methods and materials used for the degradation od Rhodamine B. For instance, titanium dioxide (TiO2) nanomaterials are a very common material used for this end.
In general, I feel the subject has significant importance and this manuscript should be recommended for publication after revising these points.
Author Response
Response to Reviewers
Reviewer 1:
Comments and Suggestions for Authors:
The manuscript submitted to Catalysts entitled "Enhancing Photocatalytic Activity of ZnO Nanoparticles in a Circulating Fluidized Bed with Plasma Jets" by Ma and co-workers presents the preparation and characterization of ZnO nanoparticles and their application in rhodamine B photodegradation. The overall subject is interesting and zinc oxide (ZnO) nanoparticles have emerged in recent years as a cost effective route for the treatment of organic pollutants and the transformation of hazardous substances into benign forms and the results of the current study, are in general interesting.
However, a few questions can be raised from the way the authors introduce and discuss the subject.
Response:Thank you very much for reviewing our manuscript. And we are really grateful for your approval of our work and helpful comments. The responses to every comment are as follows.
- The introduction seems confusing and it does not highlight the subject properly. It should be clarified why ZnO nanoparticles are a subject of interest, and for the average reader this is not clear. Furthermore, there some typos and phrases that do not add meaningful information to the introduction (e.g. line 38: "Especially, ZnO performs well on the photocatalytic.").
Response: We have revised the introduction and deleted some meaningless words (page 1).
Also, there are missing some references in the first two sentences of the introduction. I feel this subject has been discussed elsewhere and these references should be used at the end of these sentences.
Response: According to your suggestion, we have added corresponding references as listed in revised manuscript: [1] and [2,3].
- In figure 2 the authors compare the intensity of the peaks on the DRX pattern of ZnO and T4 samples and this led to the conclusion that there is less crystallinity in sample T4. If I am not mistaken the authors should compare the peak full width at half maximum of each material in order to draw any conclusion about crystallinity. This is not clear in the text neither in the figure. Perhaps the authors should provide an inset in this figure with an expansion of one peak for comparison.
Response: Under your kind suggestion, the relevant diagrams (Fig. 2) and expressions have been modified (page 4, 5).
- c) In the catalytic studies the authors show that material T5 shows less photocatalytic performance against rhodamine B when compared with material T4. However, no characterisation (PDRX, SEM, N2isotherms… etc) of material T5 is provided by the authors. Somewhere between material T4 and T5 there is a threshold where the plasma treatment changes completely the structure-affinity of these materials. In this regard, it would add value if this characterisation was available in the manuscript, since there should be major structural differences between materials T4 and T5..
Response: We are sorry for missing the important information. In the analysis of T-5 samples, we refer to the following documents:
- Dao, H.T.; Makino, H. Enhancement in optoelectrical properties of polycrystalline ZnO thin films by Ar plasma. Mat Sci Semicon Proc 2019, 96, 46-52.
- Takahashi, I.; Hayashi, Y. Synthesis of hydrogen-doped zinc oxide transparent conductive films by RF magnetron sputtering. Jan J Appl Phys2014, 54, 01AD07.
- Kim, W.; Bang, J.H.; Uhm, H.S.; Lee, S.H.; Park, J.S. Effects of post treatment on material properties and device characteristics in indium zinc oxide. Thin Solid Films2010, 519, 1573-1577.
In the description of the documents, too long plasma treatment time will cause irreversible damage to ZnO, and the descriptions in several documents are consistent.
- d) The degradation results with ZnO materials should be compared against the most common methods and materials used for the degradation od Rhodamine B. For instance, titanium dioxide (TiO2) nanomaterials are a very common material used for this end.
Response: Thank you very much for your kind suggestion. In this article, we mainly studied the effect of plasma modification on the photocatalytic effect of ZnO. Because ZnO has a lower cost than TiO2 and is more suitable for industrial batch modification, the research on TiO2 will be carried out in the future and will be compared with the results of this article.
In general, I feel the subject has significant importance and this manuscript should be recommended for publication after revising these points.
Response: Thank you for considering our manuscript. We also greatly appreciate you for the constructive comments that are helpful for improving our work. The manuscript has been revised with thorough consideration of your comments.

Reviewer 2 Report
Review of the article “Enhancing Photocatalytic Activity of ZnO Nanoparticles in a Circulating Fluidized Bed with Plasma Jets” by S. Ma, Y. Huang, R. Hong, X. Lu and J. Li, Catalysts, Manuscript ID catalysts-1049079.
In this article, the modification of ZnO nanoparticles by means of a circulating fluidized bed fed with argon and hydrogen AC arc plasma is presented. This method is used to modify the surface of ZnO nanoparticles with a view to enhance their photocatalytic activity, which is tested using Rhodamine B solutions under UV light irradiation. The authors prove that a 20 minute treatment with plasma increases the photodegradation rate constant by a factor of 6. Moreover, the modified nanoparticles exhibit an excellent reusability, maintaining a degree of degradation of circa 90% after 5 irradiation cycles. The authors claim that the improvement in the photocatalytic activity is due to an increase of specific surface area and to the derivatization of the surface with hydrogen atoms, which act as shallow donors.
Major comments:
The method used by the authors to enhance the photocatalytic activity of ZnO nanoparticles is interesting and provide good results in terms of degree of degradation of Rhodamine B. Nevertheless, some issues about the data obtained have to be addressed by the authors:
- Line 192: the authors registered relatively long lifetimes for electron-hole recombination (cf. Han, N. S. et al. Defect states of ZnO nanoparticles: Discrimination by time-resolved photoluminescence spectroscopy. J. Appl. Phys. 107, 084306, doi:10.1063/1.3382915 (2010); Fiedler, S. et al. Correlative Study of Enhanced Excitonic Emission in ZnO Coated with Al Nanoparticles using Electron and Laser Excitation. Sci. Rep. 10, 2553, doi:10.1038/s41598-020-59326-3 (2020); Appavoo, K., Liu, M. & Sfeir, M. Y. Role of size and defects in ultrafast broadband emission dynamics of ZnO nanostructures. Appl. Phys. Lett. 104, 133101, doi:10.1063/1.4868534 (2014)). How do they explain these values? And which physical meaning do these three lifetimes used to fit the decays have?
- Line 194: what is the origin of the band at 450 nm in the spectrum of T-4 in Figure 6(a)? Moreover, the pristine ZnO does not present bands in the visible range (cf. Kumar Jangir, L., Kumari, Y., Kumar, A., Kumar, M. & Awasthi, K. Investigation of luminescence and structural properties of ZnO nanoparticles, synthesized with different precursors. Mat. Chem. Front. 1, 1413-1421, doi:10.1039/C7QM00058H (2017).): how can this be explained?
- Lines 233-234: the authors have to prove this this statement via XRD and/or XPS measurements.
- Lines 315-316: the authors should clarify the meaning of this sentence.
- Line 340: the scheme for T4 sample in Figure 12 is wrong. If hydrogen atoms act as shallow donors, the relevant energy levels are located at an energy slightly lower than that of CB.
- Line 135: please specify the excitation wavelength (365 nm) of the mercury lamp.
- Line 152: please add also JCPDS standard card (No. 361451) to the figure, so that the reference peaks can be superimposed with the experimental ones.
- Lines 155-156: please insert a table with lattice constants of ZnO NPs before and after the treatment for a straightforward comparison.
- Lines 179-181: the statement is not correct, since pristine ZnO NPs present only pores at ca. 3 nm, according to Figure 5(b).
- Line 189: the authors should specify the wavelength at which the intensity is registered, i.e. 380 nm.
- Line 190: please insert references.
- Line 195: at which wavelength were the decays shown in Figure 6 b) and c) registered?
- Line 203: please insert references.
- Line 270: please insert references.
- Lines 271-272: please substitute with “an absorption ability also in the visible range”.
- Lines 273-291: there is no need for this description. The authors should delete this part.
Minor comments:
- Line 13: please write “alternating-current”
- Line 19: the photocatalytic degradation rate is ca. six times higher than that of pure ZnO, not ten (0.00396 vs 0.00065 min-1).
- Line 35: please write “binding”.
- Line 39: please substitute “on the photocatalytic” with “for photocatalytic applications”.
- Line 39: delete “the visible or”.
- Lines 43 + 287: please substitute “reducibility” with “reducing ability”.
- Lines 50-52: please write “To improve the photocatalytic performance of ZnO, the reduction of the energy band gap of ZnO and the effective separation of the photogenerated carriers have to be achieved".
- Lines 52-53: please delete the sentence from “increasing” to “ZnO”.
- Lines 66 + 69+ 71: et al. has to be in Italics.
- Line 85: please write “ions” instead of “irons”.
- Line 91: please add a space after “ZnO”
- Line 100: please delete the dot after “Table 1”.
- Line 125: please write “visible”.
- Line 134: please write “Subsequently,” instead of “Then”.
- Line 140: please write “equation 1” instead of “the formula C/C0”.
- Line 159: please write “bulk crystalline structure” instead of “inside”.
- Line 166: please write “agglomeration” instead of “reunion”.
- Line 222: please write “degradation” instead of “photolysis”.
- Line 224: please write “degradation experiment after addition of”.
- Line 225: please insert the relevant value after 100 minutes.
- Line 247: please delete the space between “photo” and “catalytic”.
- Line 309: please write “recycling”.
- Line 215: th superscript.
- Line 329: please write “result” instead of “be resulted”.
- Lines 333 + 335: please substitute “too long” with “prolonged”.
Author Response
Response to Reviewers
Reviewer 2:
Comments and Suggestions for Authors:
The method used by the authors to enhance the photocatalytic activity of ZnO nanoparticles is interesting and provide good results in terms of degree of degradation of Rhodamine B. Nevertheless, some issues about the data obtained have to be addressed by the authors
Response: Thank you very much for reviewing our manuscript. And we are really grateful for your approval of our work and helpful comments. The responses to every comment are as follows
Major comments:
- Line 192: the authors registered relatively long lifetimes for electron-hole recombination (cf. Han, N. S. et al. Defect states of ZnO nanoparticles: Discrimination by time-resolved photoluminescence spectroscopy. J. Appl. Phys. 107, 084306, doi:10.1063/1.3382915 (2010); Fiedler, S. et al. Correlative Study of Enhanced Excitonic Emission in ZnO Coated with Al Nanoparticles using Electron and Laser Excitation. Sci. Rep. 10, 2553, doi:10.1038/s41598-020-59326-3 (2020); Appavoo, K., Liu, M. & Sfeir, M. Y. Role of size and defects in ultrafast broadband emission dynamics of ZnO nanostructures. Appl. Phys. Lett. 104, 133101, doi:10.1063/1.4868534 (2014)). How do they explain these values? And which physical meaning do these three lifetimes used to fit the decays have?
Response: In the three papers given by the reviewer, the authors characterized the decay curve of fluorescence lifetime and performed multi-exponential fitting of the decay curve. The physical meaning of three lifetimes is decay times and for a single material, the decay of fluorescence lifetime is basically the same as the decay of carrier lifetime. We refer to the three documents improved by the reviewers and modify the original content.
- Line 194: what is the origin of the band at 450 nm in the spectrum of T-4 in Figure 6(a)? Moreover, the pristine ZnO does not present bands in the visible range (cf. Kumar Jangir, L., Kumari, Y., Kumar, A., Kumar, M. & Awasthi, K. Investigation of luminescence and structural properties of ZnO nanoparticles, synthesized with different precursors. Mat. Chem. Front. 1, 1413-1421, doi:10.1039/C7QM00058H (2017).): how can this be explained?
Response: According to the references you provide, the origin of the band at 420-650 nm is attributed to the existence of oxygen vacancies on the surface of ZnO. The reason why the pristine ZnO does not present bands in the visible range may be the low concentration of oxygen vacancies in pure zinc oxide. In addition, referring to this document, we also modify the original content.
- Lines 233-234: the authors have to prove this this statement via XRD and/or XPS measurements.
Response: The XRD and XPS diagrams are shown in Figure 2 and Figure 7, respectively. And regarding the principle that the photocatalytic effect will decrease due to prolonged treatment , mainly refer to the following documents:
(1). Dao, H.T.; Makino, H. Enhancement in optoelectrical properties of polycrystalline ZnO thin films by Ar plasma. Mat Sci Semicon Proc 2019, 96, 46-52.
(2). Takahashi, I.; Hayashi, Y. Synthesis of hydrogen-doped zinc oxide transparent conductive films by RF magnetron sputtering. Jan J Appl Phys 2014, 54, 01AD07.
(3). Kim, W.; Bang, J.H.; Uhm, H.S.; Lee, S.H.; Park, J.S. Effects of post treatment on material properties and device characteristics in indium zinc oxide. Thin Solid Films 2010, 519, 1573-1577.
- Lines 315-316: the authors should clarify the meaning of this sentence.
Response: The meaning of this sentence is to express that the photocatalytic effect is reduced due to the carrier scattering caused by the increase of impurity ions in the cycle experiment.
- Line 340: the scheme for T4 sample in Figure 12 is wrong. If hydrogen atoms act as shallow donors, the relevant energy levels are located at an energy slightly lower than that of CB.
Response: We are sorry for missing the important information, Figure. 12 has been revised.
- Line 135: please specify the excitation wavelength (365 nm) of the mercury lamp.
Response: The wavelength has been added.
- Line 152: please add also JCPDS standard card (No. 361451) to the figure, so that the reference peaks can be superimposed with the experimental ones.
Response: Since this standard table is in the reference, there may not be very detailed data. For details, please refer to the following documents:
(1). Omri, K.; Bettaibi, A.; Khirouni, K.; El Mir, L. The optoelectronic properties and role of Cu concentration on the structural and electrical properties of Cu doped ZnO nanoparticles. Physica B 2018, 537, 167-175.
- Lines 155-156: please insert a table with lattice constants of ZnO NPs before and after the treatment for a straightforward comparison.
Response: According to the calculation formula of the lattice constant, the lattice constant is related to 2-Theta. The change of lattice constant can be observed qualitatively according to the red shift or blue shift of the image.
- Lines 179-181: the statement is not correct, since pristine ZnO NPs present only pores at ca. 3 nm, according to Figure 5(b).
Response: We are sorry for missing the important information, the error message has been modified.
- Line 189: the authors should specify the wavelength at which the intensity is registered, i.e. 380 nm.
Response: The word has been revised.
- Line 190: please insert references.
Response: Thanks for your kind suggestion, the references has been inserted as listed in revised manuscript: [37] and [39].
- Line 195: at which wavelength were the decays shown in Figure 6 b) and c) registered?
Response: We are sorry for missing the important information, the wavelength is 375 nm.
- Line 203: please insert references.
Response: The references has been inserted as listed in revised manuscript: [43].
- Line 270: please insert references.
Response: The references has been inserted as listed in revised manuscript: [44].
- Lines 271-272: please substitute with “an absorption ability also in the visible range”.
Response: Thanks for your kind suggestion, the sentence has been substituted.
- Lines 273-291: there is no need for this description. The authors should delete this part.
Response: We have made deletions and modifications to these two paragraphs
Minor comments:
- Line 13: please write “alternating-current”
Response: Under your kind suggestion, this issue has been revised
2.Line 19: the photocatalytic degradation rate is ca. six times higher than that of pure ZnO, not ten (0.00396 vs 0.00065 min-1).
Response: According to Table. 2, the rate of T-4 is 22 times than that of pure zinc oxide (0.00396 vs 0.08933 min-1).
- Line 35: please write “binding”.
Response: This issue has been revised.
- Line39: please substitute “on the photocatalytic” with “for photocatalytic applications”.
Response: Thanks for your kind suggestion, we deleted this sentence and added a paragraph considering the meaning of improvement.
- Line 39: delete “the visible or”.
Response: Under your kind advice, the words have been deleted.
- Lines 43 + 287: please substitute “reducibility” with “reducing ability”.
Response: The word has been revised.
- Lines 50-52: please write “To improve the photocatalytic performance of ZnO, the reduction of the energy band gap of ZnO and the effective separation of the photogenerated carriers have to be achieved".
Response: The sentence has been modified.
- Lines 52-53: please delete the sentence from “increasing” to “ZnO”.
Response: The sentence has been deleted.
- Lines 66 + 69+ 71: et al. has to be in Italics.
Response: Thanks for your kind suggestion, format error has been corrected.
- Line 85: please write “ions” instead of “irons”.
Response: Under your kind advice, the word has been revised.
- Line 91: please add a space after “ZnO”
Response: The space has been added.
- Line 100: please delete the dot after “Table 1”.
Response: The dot after “Table 1” has been deleted.
- Line 125: please write “visible”.
Response: The word has been revised.
- Line 134: please write “Subsequently,” instead of “Then”.
Response:The word has been revised.
- Line 140: please write “equation 1” instead of “the formula C/C0”.
Response: The words have been revised.
- Line 159: please write “bulk crystalline structure” instead of “inside”.
Response: The words have been revised.
- Line 166: please write “agglomeration” instead of “reunion”.
Response: The word has been revised.
- Line 222: please write “degradation” instead of “photolysis”.
Response: The word has been revised.
- Line 224: please write “degradation experiment after addition of”.
Response: The words hav been added.
- Line 225: please insert the relevant value after 100 minutes.
Response: The value has been added.
- Line 247: please delete the space between “photo” and “catalytic”.
Response: The space has been deleted
- Line 309: please write “recycling”.
Response: The word has been revised.
- Line 315: th superscript.
Response: The superscript is added.
- Line 329: please write “result” instead of “be resulted”.
Response: The word has been revised.
- Lines 333 + 335: please substitute “too long” with “prolonged”.
Response: Under your kind advice, the words have been revised.

Reviewer 3 Report
Authors synthesized zinc oxide (ZnO) nanoparticles by a developed technique and tested a photocatalytic activity for RhB. This work can be published, but more experimental data should be included in the paper, and they should discuss the results. They showed photocatalytic degradation kinetics for pure ZnO, ZnO (T-1, T-2, 218 T-3. T-4, T-5), but the XPS, UV-Vis, PL, lifetime, and XRD were given only for ZnO and T-4. The data for T-1, T-2, T-3 and T-5 samples are missing. Additionally, methyl orange and methylene blue should be tested if the conclusion is same.
Author Response
Response to Reviewers
Reviewer 3:
Comments and Suggestions for Authors:
Authors synthesized zinc oxide (ZnO) nanoparticles by a developed technique and tested a photocatalytic activity for RhB. This work can be published.
Response:Thank you very much for reviewing our manuscript. And we are really grateful for your approval of our work and helpful comments. The responses to every comment are as follows.
- They showed photocatalytic degradation kinetics for pure ZnO, ZnO (T-1, T-2, T-3,T-4, T-5), but the XPS, UV-Vis, PL, lifetime, and XRD were given only for ZnO and T-4. The data for T-1, T-2, T-3 and T-5 samples are missing.
Response: The reason for only pure and T-4 samples are: (1) the T-4 sample has the best photocatalytic effect compared with the original pure ZnO, the difference can be seen to analyze the plasma modification mechanism; (2) the test characterization chart of T-2, T-3 and T-5 is very similar. In order to better display the results of the plasma processor, their corresponding data chart is not added.
- Additionally, methyl orange and methylene blue should be tested if the conclusion is same.
Response: Thank you very much for your kind suggestion. The research on methyl orange and methylene blue was studied and we refer to the following documents:
- Nam, S.H.; Boo, J.H. Enhancement of photocatalytic activity of synthesized ZnO nanoparticles with oxygen plasma treatment. Catal Today 2016, 265, 84-89.
- Savastenko, K.; Filatov, I.; Lyushkevish, V.; Chubrik, N.; Gabdullin, M.; Ramazanov, T.; Abdullin, H.; Kalkozova, V. Enhancement of ZnO-based photocatalyst activity by RF discharge-plasma treatment. J Appl Spectrosc2016, 83, 757-763.

Reviewer 4 Report
This manuscript reports the modification of zinc oxide (ZnO) nanoparticles in a circulating fluidized bed through argon and hydrogen (Ar-H) alternative-current (AC) arc plasma. The photocatalytic activity of ZnO before and after modification was tested on Rhodamine B (RhB). ZnO after 20 minutes-treatment showed ten times greater photocatalytic degradation rate than that of pure ZnO. The manuscripts can be considered for publication after addressing the following points:
- In the abstract, the authors said, “ZnO after 20 minutes-treatment by Ar-H plasma showed RhB photocatalytic degradation rate is ten times greater than that of pure ZnO”, but in the manuscript, there is no data about the ZnO after 20 min-treatment.
- In lines 165-167, the authors mentioned that “After 120 minutes of the modification process, the etching and bombardment effects were found on the surface of ZnO as well as a decrease in the degree of reunion, accompanied by smoother surfaces and smaller roughness”. As seen in Figure 4, after 120 min plasma treatment, the surface of NPs reduced roughness and became smoother, but the degree of NPs reunion is high. The authors should clarify this issue by taking high-resolution images. Additionally, scale bars are not clear and difficult to see them.
- In lines 175-176, the authors said the specific surface area of ZnO increased to 20.5630 m2/g after 120 minutes of plasma treatment, but in Figure 5(a), the results showed 4.7856 m2/g surface area of T-4 sample. The figure and description are opposite to each other; authors should verify the results.
- In section 3.4, the authors indicate the presence of OH radicals from O-1s spectra, the peak at 532.7 eV could also be generated from the surface hydroxyl group. The effect of hydroxyl radical is different from the hydroxyl group. The authors can do some experiments to confirm the presence of hydroxyl radicals.
- The authors showed that their product has low crystallinity but high photocatalytic activity. It would be better that the authors explain these results elaborately with proper references.
- The plasma modification process did not affect the change of the main absorption band of ZnO, and the authors suggest that the increased photocatalytic activity is due to modification of surface (increase surface area), the introduction of H atoms/ions, and hydroxyl radical, presence of oxygen vacancies. The authors need to prove experimentally or theoretically (with suitable reference) the existence of H atoms/ions.
- The present work is focused on the surface treatment. Also, the surface morphology is an important factor for the photocatalytic activity. Thus, the percentage of RhB removal in the dark condition should be presented because the adsorption may have occurred on the surface.
- The authors provide energy band theory for the catalytic process of ZnO. In lines 287-288, it said that the “photogenerated holes have extremely strong oxidizing properties. They will react with water and oxygen on the surface of ZnO to generate various free radicals, such as: OH, O2·-, etc.” The conversion of water to hydroxyl radical requires high potential. The energy of VB potential should be enough to execute the reaction. The authors need to evaluate the VB and CB edge potential.
Author Response
Response to Reviewers
Reviewer 4:
Comments and Suggestions for Authors:
This manuscript reports the modification of zinc oxide (ZnO) nanoparticles in a circulating fluidized bed through argon and hydrogen (Ar-H) alternative-current (AC) arc plasma. The photocatalytic activity of ZnO before and after modification was tested on Rhodamine B (RhB). ZnO after 20 minutes-treatment showed ten times greater photocatalytic degradation rate than that of pure ZnO. The manuscripts can be considered for publication after addressing the following points:.
Response: Thank you very much for reviewing our manuscript. And we are really grateful for your approval of our work and helpful comments. The responses to every comment are as follows.
- In the abstract, the authors said, “ZnO after 20 minutes-treatment by Ar-H plasma showed RhB photocatalytic degradation rate is ten times greater than that of pure ZnO”, but in the manuscript, there is no data about the ZnO after 20 min-treatment.
Response: We are sorry for such a writing error, the original text has been changed to "120min".
- In lines 165-167, the authors mentioned that “After 120 minutes of the modification process, the etching and bombardment effects were found on the surface of ZnO as well as a decrease in the degree of reunion, accompanied by smoother surfaces and smaller roughness”. As seen in Figure 4, after 120 min plasma treatment, the surface of NPs reduced roughness and became smoother, but the degree of NPs reunion is high. The authors should clarify this issue by taking high-resolution images. Additionally, scale bars are not clear and difficult to see them.
Response: Scale bars are revised in Fig. 2. The decrease in the degree of agglomeration is obtained by analyzing the BET data and the SEM picture, because the specific surface area of the sphere is the largest, and the spherical structure is destroyed after the plasma treatment, but the specific surface area is significantly increased due to the decrease in the degree of agglomeration.
- In lines 175-176, the authors said the specific surface area of ZnO increased to 20.5630 m2/g after 120 minutes of plasma treatment, but in Figure 5(a), the results showed 4.7856 m2/g surface area of T-4 sample. The figure and description are opposite to each other; authors should verify the results.
Response: Under your kind suggestion, Figure. 5 has been modified.
- In section 3.4, the authors indicate the presence of OH radicals from O-1s spectra, the peak at 532.7 eV could also be generated from the surface hydroxyl group. The effect of hydroxyl radical is different from the hydroxyl group. The authors can do some experiments to confirm the presence of hydroxyl radicals.
Response: We are sorry for the wrong expression. Because XPS analysis is to perform elemental analysis on the surface of the sample, the peak at 533.7eV should correspond to plasma treatment which can enhance the physical and chemical adsorption of hydroxyl radicals on the surface of ZnO..
- The authors showed that their product has low crystallinity but high photocatalytic activity. It would be better that the authors explain these results elaborately with proper references.
Response: Plasma treatment introduces new ions/atoms at the grain boundaries, and the lattice units in the crystal grains are deformed and subjected to stress which resulting in a decrease in the degree of crystallinity, but at the same time the introduced H acts as a shallow layer in ZnO, thereby enhancing the photocatalytic ability of zinc oxide. Mainly refer to the following documents:
[1] Dao, H.T.; Makino, H. Enhancement in optoelectrical properties of polycrystalline
ZnO thin films by Ar plasma. Mat Sci Semicon Proc 2019, 96, 46-52.
- The plasma modification process did not affect the change of the main absorption band of ZnO, and the authors suggest that the increased photocatalytic activity is due to modification of surface (increase surface area), the introduction of H atoms/ions, and hydroxyl radical, presence of oxygen vacancies. The authors need to prove experimentally or theoretically (with suitable reference) the existence of H atoms/ions.
Response: Regarding the problems of H atoms/ions, we mainly refer to the following two documents:
- Ohashi, N.; Wang, Y.G.; Ishigaki, T.; Wada, Y.; Taguchi, H.; Sakaguchi, I.; Ohgaki, T.; Adachi, Y. Low stimulated emission threshold of zinc oxide by hydrogen doping with pulsed argon-hydrogen plasma.J Cryst Growth 2007, 306, 316-320.
- Dev, A.; Niepelt, R.; Richters, J.P.; Ronning, C.; Voss, T. Stable enhancement of near-band-edge emission of ZnO nanowires by hydrogen incorporation.Nanotechnology 2015, 21, 065709.
- The present work is focused on the surface treatment. Also, the surface morphology is an important factor for the photocatalytic activity. Thus, the percentage of RhB removal in the dark condition should be presented because the adsorption may have occurred on the surface.
Response: The experiment under dark conditions is shown in the curve of “Blank” in Fig. 8.
- The authors provide energy band theory for the catalytic process of ZnO. In lines 287-288, it said that the “photogenerated holes have extremely strong oxidizing properties. They will react with water and oxygen on the surface of ZnO to generate various free radicals, such as: OH, O2·-, etc.” The conversion of water to hydroxyl radical requires high potential. The energy of VB potential should be enough to execute the reaction. The authors need to evaluate the VB and CB edge potential.
Response: Although the academic circles have not yet concluded which active group plays a leading role, the photocatalytic mechanism of zinc oxide has been reported in many literatures. We summarized the report on the photocatalytic mechanism of ZnO based on the following literature:
[1] Sivakarthik, P.; Thangaraj, V.; Parthibavarman, M. A facile and one-pot synthesis of pure and transition metals (M=Co&Ni) doped WO3 nanoparticles for enhanced photocatalytic performance. J Mater Sci-Mater El 2017, 28, 5990-5996.

Round 2
Reviewer 1 Report
In general, I am pleased with the author's responses to my concerns, and the general improvements made to the manuscript after the initial assessment. I am happy to recommend the manuscript for publication in the current form.
Author Response
Reviewer 1:
Comments and Suggestions for Authors:
In general, I am pleased with the author's responses to my concerns, and the general improvements made to the manuscript after the initial assessment. I am happy to recommend the manuscript for publication in the current form.
Response: Thank you very much for your valuable comments, which are of great help to the revision of the manuscript. We sincerely thank you for your precious comments.

Reviewer 2 Report
Second review of the article “Enhancing Photocatalytic Activity of ZnO Nanoparticles in a Circulating Fluidized Bed with Plasma Jets” by S. Ma, Y. Huang, R. Hong, X. Lu and J. Li, Catalysts, Manuscript ID catalysts-1049079.
The authors replied satisfactorily only to a part of the questions.
- As for Question 1 (Major comments), the authors still do not explain the physical meaning i.e. the origin of the three lifetimes obtained from the fitting of the luminescence decay of their materials. It is common knowledge that the same semiconductor material can present different exciton lifetimes depending on the different recombination mechanisms (see for example Kahn, M. L. et al. Optical Properties of Zinc Oxide Nanoparticles and Nanorods Synthesized Using an Organometallic Method. ChemPhysChem 7, 2392-2397, doi:https://doi.org/10.1002/cphc.200600184 (2006)). I strongly suggest the authors to limit the discussion to the average lifetimes τavg (calculated as below for a triple exponential decay; ref. Lakowicz, J. R. (2006). Principles of fluorescence spectroscopy. New York: Springer., page 142) and to compare the values before and after plasma treatment. This means that the discussion about τ1, τ2, τ3 have to be deleted in the text.
τavg = (α1τ12 + α2τ22 + α3τ32)/(α1τ1 + α2τ2 + α3τ3)
- Line 97: the calculated values of full width at half maximum (FWHM) for pure ZnO and T-4 have to be indicated in the text.
- Line 97: the decrease in FWHM does not indicate a decrease in crystallinity, but an increase in the crystallite size, according to Scherrer equation.
- Line 142-148: move these lines to the experimental part and discuss only the average lifetimes.
- Line 146-148: please delete the statement “For a single material, the decay of fluorescence lifetime is basically the same as the decay of carrier”, because is not properly true.
- Line 148: the proper terminology is “average lifetime”, not “compound lifetime”.
- Line 151: please delete the values of , , in Figure 5 b) and c).
- Line 199: please use “absorbance” instead of “light absorption intensity”.
- Line 208-209: please write “for this experiment a fitting analysis was performed up to a 90% degree of degradation”.
- Line 230-231: please change to “ZnO exhibits a valence band and a conduction band”.
- Line 232-234: the authors should rephrase this part with a more scientific language.
Minor comments:
- Line 38: please write “semiconductors”.
- Line 137: please write “luminescence intensity”.
- Line 137-141: the concept is repeated twice.
- Line 182: please delete “adding”.
- Line 309: please write “the suspension was kept in dark conditions for 60 mins…”.
- Line 313: please add “spectrophotometer” between “UV-vis” and “was”.
- Line 313: please change to “to analyse”.
- Line 326: please substitute “Then” with “Moreover”.
- Line 348: please substitute “summation” with “taking into consideration”.
- Line 349: please substitute “and so on” with “other techniques”.
Author Response
Response to Reviewers
Reviewer 2:
Comments and Suggestions for Authors:
Second review of the article “Enhancing Photocatalytic Activity of ZnO Nanoparticles in a Circulating Fluidized Bed with Plasma Jets” by S. Ma, Y. Huang, R. Hong, X. Lu and J. Li, Catalysts, Manuscript ID catalysts-1049079. The authors replied satisfactorily only to a part of the questions.
Response: Thank you very much for reviewing our manuscript. We are sorry for not fully answering your question satisfactorily. And we are really grateful for your approval of our work and helpful comments. The responses to every comment are as follows
Major comments:
- As for Question 1 (Major comments), the authors still do not explain the physical meaning i.e. the origin of the three lifetimes obtained from the fitting of the luminescence decay of their materials. It is common knowledge that the same semiconductor material can present different exciton lifetimes depending on the different recombination mechanisms (see for example Kahn, M. L. et al. Optical Properties of Zinc Oxide Nanoparticles and Nanorods Synthesized Using an Organometallic Method.Chem Phys Chem 7, 2392-2397, doi: https: //doi.org/10.1002/cphc. 200600184 (2006)). I strongly suggest the authors to limit the discussion to the average lifetimes τavg (calculated as below for a triple exponential decay; ref. Lakowicz, J. R. (2006). Principles of fluorescence spectroscopy. New York: Springer., page 142) and to compare the values before and after plasma treatment. This means that the discussion about τ1, τ2, τ3 have to be deleted in the text.
τavg = (α1τ12 + α2τ22 + α3τ32)/(α1τ1 + α2τ2 + α3τ3)
Response: We are sorry for not understanding your meaning properly, and also sorry for not having an in-depth understanding of this knowledge. We have revised the original text and deleted the discussion about τ1, τ2, τ3, only discussing the average lifetime.
- Line 97: the calculated values of full width at half maximum (FWHM) for pure ZnO and T-4 have to be indicated in the text.
Response: We are sorry for missing the important information, the values of full width at half maximum (FWHM) for pure ZnO and T-4 have been indicated in the manuscript in page 3.
- Line 97: the decrease in FWHM does not indicate a decrease in crystallinity, but an increase in the crystallite size, according to Scherrer equation.
Response: According to your kind suggestion, we have modified the original content in line 97.
- Line 142-148: move these lines to the experimental part and discuss only the average lifetimes.
Response: These lines have been moved.
- Line 146-148: please delete the statement “For a single material, the decay of fluorescence lifetime is basically the same as the decay of carrier”, because is not properly true.
Response: We are sorry for missing the important information, the sentence has been deleted.
- Line 148: the proper terminology is “average lifetime”, not “compound lifetime”.
Response: The words have been revised.
- Line 151: please delete the values of , , in Figure 5 b) and c).
Response: The values have been deleted.
- Line 199: please use “absorbance” instead of “light absorption intensity”.
Response: The words have been revised.
- Line 208-209: please write “for this experiment a fitting analysis was performed up to a 90% degree of degradation”.
Response: The sentence has been revised in lines 207-208.
- Line 230-231: please change to “ZnO exhibits a valence band and a conduction band”.
Response: The sentence has been revised in line 229.
- Line 232-234: the authors should rephrase this part with a more scientific language.
Response: The sentences have been revised in lines 231-233.
Minor comments:
- Line 38: please write “semiconductors”.
Response: Under your kind suggestion, the word has been revised.
- Line 137: please write “luminescence intensity”.
Response: The words have been revised..
- Line 137-141: the concept is repeated twice.
Response: Thanks for your kind suggestion, the sentence has been deleted.
- Line 182: please delete “adding”.
Response: The word has been deleted.
- Line 309: please write “the suspension was kept in dark conditions for 60 mins…”.
Response: Under your kind advice, the sentence has been modified.
- Line 313: please add “spectrophotometer” between “UV-vis” and “was”.
Response: The word has been added.
- Line 313: please change to “to analyse”.
Response: The word has been revised.
- Line 326: please substitute “Then” with “Moreover”.
Response: The word has been substituted.
- Line 348: please substitute “summation” with “taking into consideration”.
Response: Thanks for your kind suggestion, the word has been substituted.
- Line 349: please substitute “and so on” with “other techniques”.
Response: Under your kind advice, the word has been revised.

Reviewer 3 Report
Authors revised the manuscript.
Optional comments:
Still this reviwer is asking why not show other fundamental data although they are similar.
Also, to increase impact it is better to test MO and MB with the same materials under the same experimental condition.
Author Response
Response to Reviewers
Reviewer 3:
Comments and Suggestions for Authors:
Authors revised the manuscript.
Response:Thank you very much for reviewing our manuscript. We sincerely thank you for your precious comments. And we are really grateful for your approval of our work and helpful comments.
This reviwer is asking why not show other fundamental data although they are similar. Also, to increase impact it is better to test MO and MB with the same materials under the same experimental condition.
Response: Thank you very much for your kind suggestion. We will continue to study the plasma modified ZnO on the photocatalytic effect of MB and MO in our future work.

Reviewer 4 Report
Revision is satisfactory.
Author Response
Reviewer 4:
Comments and Suggestions for Authors:
Revision is satisfactory.
Response: Thank you very much for reviewing our manuscript. We sincerely thank you for your precious comments. And we are really grateful for your approval of our work and helpful comments.

Round 3
Reviewer 2 Report
Third review of the article “Enhancing Photocatalytic Activity of ZnO Nanoparticles in a Circulating Fluidized Bed with Plasma Jets” by S. Ma, Y. Huang, R. Hong, X. Lu and J. Li, Catalysts, Manuscript ID catalysts-1049079.
The authors replied satisfactorily to all of the questions.
Minor comments:
- Line 142: please write “profile”.
- Line 193: please substitute “absorption” with “absorbance”.
- Line 202: please insert “, therefore” before “for this experiment”.
- Line 346: please insert “and” before “other techniques”.
Author Response
Response to Reviewers
Reviewer 2:
Comments and Suggestions for Authors:
The authors replied satisfactorily to all of the questions.
Response: Thank you very much for your valuable comments, which are of great help to the revision of the manuscript. We sincerely thank you for your precious comments.
Minor comments:
- Line 142: please write “profile”.
Response: Under your kind suggestion, the word has been revised in line 142.
- Line 193: please substitute “absorption” with “absorbance”.
Response: The word have been revised in line 194.
- Line 202: please insert “, therefore” before “for this experiment”.
Response: The words have been inserted in line 203.
- Line 346: please insert “and” before “other techniques”.
Response: Thanks for your kind suggestion, the word has been inserted in line 351.
